# Osteoclast-Like Cells in Aneurysmal Disease Exhibit an Enhanced Proteolytic Phenotype

**DOI:** 10.3390/ijms20194689

**Published:** 2019-09-21

**Authors:** Matthew J. Kelly, Kimihiro Igari, Dai Yamanouchi

**Affiliations:** 1Division of Vascular Surgery, Department of Surgery, School of Medicine and Public Health, University of Wisconsin-Madison, Madison, WI 53705, USA; 2Division of Vascular and Endovascular Surgery, Department of Surgery, Tokyo Medical and Dental University (TMDU), Tokyo 113-8510, Japan

**Keywords:** aneurysm, osteoclast-like cell (OLC), hypoxia-inducible factor-1α (HIF-1α)

## Abstract

Abdominal aortic aneurysm (AAA) is among the top 20 causes of death in the United States. Surgical repair is the gold standard for AAA treatment, therefore, there is a need for non-invasive therapeutic interventions. Aneurysms are more closely associated with the osteoclast-like catabolic degradation of the artery, rather than the osteoblast-like anabolic processes of arterial calcification. We have reported the presence of osteoclast-like cells (OLCs) in human and mouse aneurysmal tissues. The aim of this study was to examine OLCs from aneurysmal tissues as a source of degenerative proteases. Aneurysmal and control tissues from humans, and from the mouse CaPO_4_ and angiotensin II (AngII) disease models, were analyzed via flow cytometry and immunofluorescence for the expression of osteoclast markers. We found higher expression of the osteoclast markers tartrate-resistant acid phosphatase (TRAP), matrix metalloproteinase-9 (MMP-9), and cathepsin K, and the signaling molecule, hypoxia-inducible factor-1α (HIF-1α), in aneurysmal tissue compared to controls. Aneurysmal tissues also contained more OLCs than controls. Additionally, more OLCs from aneurysms express HIF-1α, and produce more MMP-9 and cathepsin K, than myeloid cells from the same tissue. These data indicate that OLCs are a significant source of proteases known to be involved in aortic degradation, in which the HIF-1α signaling pathway may play an important role. Our findings suggest that OLCs may be an attractive target for non-surgical suppression of aneurysm formation due to their expression of degradative proteases.

## 1. Introduction

There are an estimated 1.1 million abdominal aortic aneurysms (AAAs) in the United States according to a comprehensive analysis of risk factors for AAA in more than three million individuals evaluated by ultrasound imaging [1]. In 2015, AAA was associated with 16,522 deaths in the US, and this number is expected to grow due to an aging population demographic [2]. The risk factors for the development of AAA include age, male gender, family history, cardiovascular disease, and smoking [1,3,4]. Unless detected via imaging scans, often for other reasons, AAAs frequently remain undiagnosed until rupture. Rupture is a highly fatal event. About half of patients survive long enough to make it to surgery, and of those that do, the mortality rate approaches 50% [5]. Existing guidelines suggest the option of elective surgical repair for aneurysms >5–5.5 cm in diameter [6]. Currently, open surgical repair and endovascular placement of a stent graft are the only proven treatments for AAA [7,8]. The significant invasiveness and morbidity and mortality associated with surgery stresses the need for alternative non-surgical therapeutic strategies [9,10].

The areas of aneurysmal vessels with less calcification may be the most likely sites of rupture, and vessels with greater calcification exhibit reduced aneurysm growth [11,12]. Furthermore, subsequent studies demonstrated the involvement of balanced mineralization in diseased arteries through the tight control of calcification by osteoblast-like cells, and decalcification by osteoclast-like cells (OLCs) [13,14]. OLCs are similar to osteoclasts but occur in tissues other than bone, differentiate from monocytes/macrophages, and are positive for tartrate-resistant acid phosphatase (TRAP) staining [15]. In the bone, receptor activator of nuclear factor-kappa B (NF-κB) ligand (RANKL) is the factor responsible for osteoclastogenic differentiation of macrophages. RANKL is highly expressed in the bone and is essential for the formation of mature osteoclasts, which express proteases such as matrix metalloproteinase-9 (MMP-9) and cathepsin K, and facilitate the migration of osteoclasts to resorption sites through the extracellular matrix [16,17]. We previously demonstrated the ability of both RANKL and tumor necrosis factor α (TNFα) plus CaPO_4_ to stimulate osteoclastogenic differentiation of macrophages in vitro, indicating an additional pathway (TNFα + CaPO_4_) through which macrophages can be stimulated to differentiate into osteoclasts [18]. The classical RANK/RANKL stimulation pathway activates TNF receptor-associated factor 6 (TRAF6) and calcium signaling, leading to downstream activation of NF-κB, mitogen-activated protein kinases (MAPKs), and the induction of nuclear factor of activated T cells cytoplasmic 1 (NFATc1) [19,20,21]. However, TNFα + CaPO_4_ osteoclastogenesis is mediated by TRAF2, not TRAF6 [18].

We found that osteoclastogenesis plays an important role in the development of aneurysms through stimulation of TRAP-positive macrophages (TPMs) in the CaPO_4_ and angiotensin II (AngII)-infused apolipoprotein E-deficient (apoE^−/−^) mouse models [18,22]. Medial accumulation of macrophages in mouse (CaPO_4_ and AngII-induced) and human aneurysms are a hallmark of disease progression [23], and we demonstrated the presence of TPMs in mouse and human aneurysmal tissues [18,22]. Previous studies point to the infiltration of macrophages from the luminal side of the vessel, concentrated in areas covered by thin thrombus, as important sources of degradative proteases, like MMP-9 [24,25,26]. The vital role of MMP-9 in aneurysmal disease has long been recognized [27,28], and our previous studies corroborated the increased expression of MMP-9 in aneurysmal tissues [18,22]. Importantly, we demonstrated the differential effectiveness of two classes of osteoclast inhibitors in the CaPO_4_ and AngII mouse models of disease. Osteoclast-targeted treatment of mouse CaPO_4_-induced aneurysms (TNFα and CaPO_4_-dependent, RANKL-independent) with bisphosphonate was effective in inhibiting aneurysm formation and TRAP expression [18]. Upon treatment of AngII-induced aneurysms (TNFα and CaPO_4_-independent, RANKL-dependent) with RANKL-neutralizing antibody, we observed decreased TRAP and MMP-9 expression [22]. In light of our previous results, TPMs appear to play an important role in aneurysm formation, however, we do not know if this specific cell population is responsible for the production of vessel-degrading proteases in mouse and human aneurysmal tissues.

Hypoxia-inducible factor-1α (HIF-1α) is a transcription factor involved in many cellular processes, such as the regulation of oxygen homeostasis, metabolism, cell proliferation, and survival [29]. The specific role of HIF-1α in aneurysm formation remains unclear. However, increased HIF-1α expression has been demonstrated in aneurysmal tissues [30,31], and previous studies found that HIF-1α inhibition attenuates the progression of experimental AAAs [32,33]. Hypoxic conditions, such as those found in the aneurysmal wall, can stimulate RANK/RANKL osteoclastogenesis through induction of HIF-1α [34]. We found that RANKL-induced osteoclastogenesis of macrophages in vitro increased HIF-1α expression, and that inhibition of HIF-1α via treatment with digoxin blocked osteoclastogenesis [35]. Given these previous findings, we suspect that HIF-1α may be overexpressed in TPMs found in aneurysmal tissue.

In this study, we hypothesize that TPMs are a significant source of the degradative proteases MMP-9 and cathepsin K, like osteoclasts, and their activation may be mediated through the HIF-1α pathway. The central hypothesis of our research is that osteoclastogenic activation of macrophages contributes to aneurysmal degeneration, and thus, OLCs represent a potential target for therapeutic intervention in aneurysmal disease.

## 2. Results

### 2.1. TRAP-Positive Macrophages Produce More Cathepsin K and MMP-9 than TRAP-Negative Macrophages

We previously demonstrated that treatment of the RAW 264.7 mouse macrophage cell line with CaPO_4_ and TNFα can induce osteoclastogenesis, as indicated by increased TRAP, cathepsin K, and MMP-9 expression [18]. Next, we were interested in examining the differential expression of the markers of osteoclastogenesis within the stimulated cell population. We applied flow cytometry to further analyze the in vitro osteoclastogenic stimulation of RAW 264.7 cells, allowing us to examine protease expression in subsets of the stimulated population through the gating strategy depicted in Figure 1.

In vitro stimulation of macrophages with CaPO_4_ and TNFα results in increased TRAP (0.9633% ± 0.008819% vs. 7.733% ± 0.1556%, *p* < 0.001), cathepsin K (0.2233% + 0.04702% vs. 3.913% + 0.07364%, *p* < 0.0001, and MMP-9 (0.4267% ± 0.1415% vs. 4.057% ± 0.1648%, *p* < 0.0001) expression (Figure 2A–C). We also examined the expression of HIF-1α for its potential role in osteoclastogenic signaling and found elevated expression in stimulated cells compared to control (0.2333% ± 0.05487% vs. 1.207% ± 0.1328%, *p* < 0.01) (Figure 2D). Furthermore, among stimulated cells, we demonstrated that osteoclasts (TRAP-positive) produced more cathepsin K (5121 ± 277.6 vs. 17185 ± 513, *p* < 0.0001) and MMP-9 (6291 ± 617.3 vs. 20702 ± 961.5, *p* < 0.001) than nonactivated macrophages (TRAP-negative) (Figure 2E,F, respectively).

### 2.2. Aneurysmal Tissues Exhibit Increased Expression of Osteoclastogenic Markers

Mice treated via the CaPO_4_ and AngII models exhibited aneurysm formation in the carotid arteries and abdominal aortae, respectively, as defined by a 50% or greater increase in vessel diameter (Figure 3A,B, respectively). The treatment of carotid arteries in the CaPO_4_ model resulted in increased vessel diameter compared to untreated contralateral carotids (0.508 ± 0.01319 mm vs. 1.018 ± 0.03597 mm, *p* < 0.0001) (Figure 3C). Similarly, we found increases in the diameters of abdominal aortae upon administration of AngII compared to PBS-only controls (0.9267 ± 0.01764 mm vs. 1.942 ± 0.2552 mm, *p* < 0.05) (Figure 3D).

Mouse and human aneurysmal and control tissues were subjected to enzymatic degradation to obtain single cell suspensions, which were then stained for cell viability, CD11b, TRAP, MMP-9, and cathepsin K, and analyzed via flow cytometry according to the strategy depicted in Figure 1. The data are presented as the percent of live cells that are positive for the markers of interest. In the mouse CaPO_4_-induced aneurysm model, aneurysmal tissues expressed significantly higher levels of CD11b (0.0416% ± 0.004354% vs. 2.492% + 0.5137%, *p* < 0.01), TRAP (0.3% ± 0.1679% vs. 2.008% ± 0.2024%, *p* < 0.001), cathepsin K (0.272% ± 0.03169% vs. 2.72% ± 0.3362%, *p* < 0.01), and MMP-9 (0.202% ± 0.03247% vs. 3.02% ± 0.33%, *p* < 0.001) compared to untreated contralateral carotid artery controls (Figure 4A). Aneurysmal tissues from the mouse AngII-induced model also demonstrated a significant increase in expression of CD11b (6.507% ± 1.736% vs. 12.4% ± 1.167%, *p* < 0.05), TRAP (0.8633% ± 0.2356% vs. 2.812% ± 0.3442%, *p* < 0.01), cathepsin K (1.58% ± 0.2773% vs. 7.05% ± 2.008%, *p* < 0.05) and MMP-9 (0.51% ± 0.1301% vs. 10.02% ± 1.159%, *p* < 0.001) compared to control aortae from PBS-infused mice (Figure 4B). We also examined human carotid plaque and AAA tissues for the expression of osteoclastogenic markers, and found significant differences between human AAAs and carotid plaques in expression of CD11b (1.89% + 1.79% vs. 10.68% + 2.502%, *p* < 0.05), TRAP (0.4433% ± 0.2492% vs. 5.538% ± 1.234%, *p* < 0.05), cathepsin K (0.7787% ± 0.5116% vs. 20.03% ± 4.272%, *p* < 0.05), and MMP-9 (0.86% ± 0.5514% vs. 18.64% ± 4.068%, *p* < 0.05) (Figure 4C).

### 2.3. HIF-1α Expression is Enhanced in Aneurysmal Tissues and TPMs

In addition to the aforementioned markers of osteoclastogenesis, we were interested in evaluating expression of the transcription factor HIF-1α, as it may play an important role in osteoclastogenesis [36]. Previously, HIF-1α has been described as a potentially important pathway in the development of aneurysms in the mouse AngII model and in human tissue, but it has not been described in the mouse CaPO_4_ model [30,32,37]. We evaluated the percentage of live cells positive for HIF-1α via flow cytometry, and compared to controls, we observed increased HIF-1α expression in CaPO_4_ (0.1294% ± 0.01719% vs. 1.774% ± 0.2277%, *p* < 0.01) (Figure 5A) and AngII (0.7133% ± 0.08819% vs. 3.703% ± 0.5235%, *p* < 0.01) (Figure 5B) mouse models of aneurysm. Likewise, compared to human carotid plaque tissues, human AAA tissues showed elevated HIF-1α expression (0.3733% ± 0.2136% vs. 5.428% ± 1.582%, *p* < 0.05) (Figure 5C).

Additionally, we compared HIF-1α expression between myeloid and TPM cell populations from mouse and human aneurysms. The expression of TRAP is a long-established marker of osteoclastogenically activated macrophages [15]. Monocytes/macrophages are derived via the myeloid lineage, and CD11b is a reliable marker for these cells, which are the precursors to osteoclasts. We observed increased HIF-1α expression in TPMs from mouse CaPO_4_-induced aneurysms (21.73% ± 6.202% vs. 67.7% ± 2.195%, *p* < 0.001), AngII-induced aneurysms (5.572% ± 1.356% vs. 56.22% ± 5.458%, *p* < 0.001), and human AAAs (20% ± 4.952% vs. 82.8% ± 11.58%, *p* < 0.01) (Figure 5D–F, respectively).

### 2.4. Aneurysmal Tissues Exhibit Increased Infiltration of TPMs

We evaluated aneurysmal tissues for the presence of TPMs via flow cytometry and immunofluorescence. In the mouse CaPO_4_-induced aneurysm model, we found that aneurysmal tissues expressed significantly higher levels of TPMs compared to controls as demonstrated by flow cytometry (0.01043% ± 0.004556% vs. 1.145% ± 0.1756%, *p* < 0.01) (Figure 6A), and via immunofluorescence staining of control (Figure 6D) and aneurysmal (Figure 6G) tissue sections. Of note, a distinctly multinucleate TPM is indicated by the leftmost arrow of Figure 6G. Mouse AngII-induced aneurysms also exhibited significantly higher levels of TPMs compared to control aortae (0.19% ± 0.01732% vs. 1.125% ± 0.1194%, *p* < 0.001) via flow cytometry (Figure 6B), and immunofluorescence of control (Figure 6E) and aneurysmal sections (Figure 6H). Finally, we analyzed human carotid plaque and AAA tissues, and found an increase in the number of TPMs in AAAs compared to carotid plaques as evaluated by flow cytometry (0.02036% ± 0.01515% vs. 4.58% ± 0.4816%, *p* < 0.01) (Figure 6C), and immunofluorescence staining of carotid plaque (Figure 6F) and AAA tissue sections (Figure 6I).

### 2.5. TPMs From Aneurysmal Tissues Express Elevated Levels of Cathepsin K and MMP-9

We hypothesized that TPMs found in aneurysmal tissues contribute to vessel degradation through elevated production of proteases such as MMP-9 and cathepsin K. Consequently, we compared the median fluorescence intensity (MFI) of MMP-9 and cathepsin K in myeloid (CD11b^+^, TRAP^−^) and TPM (CD11b^+^, TRAP^+^) cell populations derived from mouse and human aneurysmal tissues. In the mouse CaPO_4_ model, we found that TPMs expressed significantly more cathepsin K (1324 ± 123.4 vs. 4594 ± 955.2, *p* < 0.05) (Figure 7A) and MMP-9 (1739 ± 168 vs. 6477 ± 138, *p* < 0.05) (Figure 7D) than myeloid cells. Similarly, in the mouse AngII model, TPMs expressed more cathepsin K (1331 ± 340.6 vs. 4161 ± 739.1, *p* < 0.01) (Figure 7B) and MMP-9 (2536 ± 427.2 vs. 5851 ± 465.1, *p* < 0.001) (Figure 7E) compared to myeloid cells. In human AAA tissues, we also found that TPMs, when compared to myeloid cells, expressed more cathepsin K (5675 ± 1542 vs. 33604 ± 7540, *p* < 0.05) (Figure 7C) and MMP-9 (5444 ± 1891 vs. 27830 ± 5823, *p* < 0.05) (Figure 7F).

## 3. Discussion

Nearly 25 years ago, Thompson et al. demonstrated increased production of the gelatinase, MMP-9, localized to macrophages in AAA tissues compared to those from normal and occlusive arteries [27]. Longo et al. elucidated the role of MMP-9 in aneurysm development in MMP-9 knockout (KO) mice, in which KO attenuated disease formation [28]. More specifically, they revealed that macrophage-derived MMP-9 and mesenchymal MMP-2 are both required for aneurysm formation in the mouse elastase model. In our previous work, we identified the presence of OLCs in aneurysmal disease and demonstrated a concurrent increase in MMP-9 production, which corroborates previous evidence of the importance of MMP-9 in AAA formation [18,22,28]. We also demonstrated that targeting of OLCs via administration of bisphosphonates, which are commonly used to treat osteoporosis, or RANKL-neutralizing antibody, decreased aneurysmal dilation in the mouse CaPO_4_ and AngII models, respectively [18,22]. In addition to MMP-9, cathepsin K has been identified as another important protease upregulated in AAA [38]. Expression of cathepsin K, a strong elastase which is highly expressed in osteoclasts and vital for their function in bone degradation, was of particular interest in our hypothesis of OLC involvement in aneurysm formation [39,40,41,42]. Cathepsin K deficiency was shown by Sun et al. to result in decreased aneurysm formation in the elastase-induced mouse model [43]. However, the AngII-induced aneurysm model, in apoE^−/−^, cathepsin K^−/−^ double KO mice, showed no attenuation of aneurysm formation [44]. A possible explanation for this unexpected result could be the compensatory action of other elastinolytic cathepsins, such as S and L, as described previously [45]. Furthermore, AngII treatment of macrophages stimulated increased production of cathepsin F, which may specifically compensate for cathepsin K in AngII-induced aneurysms compared to the elastase model [46].

In this study, we sought to further investigate whether TPMs found in aneurysmal tissue are significant sources of the vessel-degrading proteases MMP-9 and cathepsin K, similar to osteoclasts. Using flow cytometry, we evaluated live cells from aneurysmal tissue for expression of TRAP, cathepsin K, and MMP-9. We chose to compare human AAA and carotid plaque tissues for these investigations due to the distinct differences in the pathology of the two diseases. Carotid plaque formation is an atherosclerotic disease which is characterized by increased calcification and narrowing of the artery, as opposed to aneurysm formation, which does not display high levels of calcification, but rather a degeneration of the arterial wall leading to vessel dilation. We expect the calcium anabolism that is characteristic of plaque formation does not involve high levels of osteoclastogenesis, which is a catabolic event, and thus characteristic of aneurysmal disease. In confirmation of our previous studies, and those of others, we found significant increases in TRAP, cathepsin K, and MMP-9 expression in mouse and human aneurysmal tissues compared to controls. In accordance with our previous studies, we demonstrated the presence of TPMs in aneurysmal tissues via flow cytometry and immunofluorescence staining. Notably, we revealed the presence of a distinctly multinucleate TPM, a characteristic morphology of osteoclasts, in a mouse CaPO_4_-induced aneurysm (Figure 6G). Moreover, we showed that TPMs produce significantly more cathepsin K and MMP-9 than TRAP-negative myeloid cells. We demonstrated that TPMs, specifically, are producing significant amounts of proteases, and thus, functioning as OLCs. To our knowledge, this study is the first to examine discrete live cell populations from aneurysmal tissues for their differential expression of osteoclast-associated proteins.

The HIF-1α signaling pathway is involved in osteoclastogenesis and may play an important role in upregulating the proteolytic capacity of osteoclasts [36,47,48,49]. Therefore, we were particularly interested in the role of HIF-1α signaling as it relates to the potential stimulation of OLCs in aneurysmal tissue. Importantly, as we demonstrated here, OLCs in aneurysmal tissue are a significant source of the degradative proteases cathepsin K and MMP-9. In these experiments, we demonstrated increased HIF-1α expression in human and mouse aneurysmal tissues compared to controls, which may be due, in part, to hypoxic conditions found in aneurysmal tissues [31,33,50,51,52,53]. Moreover, we showed that HIF-1α expression in TPMs is significantly higher than myeloid cells. This is particularly interesting in light of the previously described role for HIF-1α in osteoclastogenesis and aneurysm formation. Further evidence of the potential importance of HIF-1α is derived from studies investigating factors related to the inhibition of HIF-1α, osteoclastogenesis, and aneurysm formation, such as female sex and diabetes. Sex is a significant risk factor for aneurysm formation, with women exhibiting approximately a 4-5-fold reduction in prevalence [54,55]. Mukundan et al. demonstrated that estradiol can inhibit hypoxia-induced HIF-1α expression in an estrogen-receptor mediated manner [56]. Likewise, estrogen deficiency served to stabilize HIF-1α in osteoclasts and lead to their activation [57]. Diabetes was identified to correlate negatively with aneurysm formation [4]. With respect to hyperglycemia, we previously showed that decreased MMP-9 expression and macrophage activation under hyperglycemic conditions correlated with suppression of aneurysm formation [58]. Subsequently, we showed that hyperglycemia suppressed macrophage activation via the downregulation of the GLUT1 receptor, which is a transcriptional target of HIF-1α [59,60]. In light of the important role of HIF-1α in osteoclast activation and aneurysm formation, in this study, we demonstrated increased HIF-1α expression in OLCs derived from aneurysmal tissues, concurrent with a significant increase in protease expression by this cell population.

The function of HIF-1α in aneurysms remains unclear, as multiple studies have demonstrated seemingly contradictory results. Previous studies have demonstrated that HIF-1α expression is negatively associated with aneurysm formation [61,62,63], while others demonstrate a positive association [32,37]. The conflicting results in these studies have not yet been sufficiently explained. However, the ambiguity with respect to the role of HIF-1α in AAA formation may be the result of variations in the models of AAA induction, or differential roles of HIF-1α with respect to temporal expression or the specific cell types investigated in these studies. In future experiments, we hope to elucidate the role of HIF-1α as it relates to aneurysm induction and progression, particularly concerning the activation and effector function of OLCs.

This study demonstrates that OLCs are resident in aneurysmal tissues. We identified these cells as significant sources of the vessel-degrading proteases cathepsin K and MMP-9, therefore, OLCs represent intriguing targets for therapeutic strategies to treat or prevent aneurysmal degeneration. However, this study has some limitations. First, we did not attempt to inhibit OLCs and evaluate tissues post-treatment for aneurysm formation, OLC infiltration, and OLC phenotype. Moreover, outside of HIF-1α expression, we did not examine the signaling pathways involved in OLC activation in aneurysmal tissues. Although previous work performed in our lab detailed the difference in signaling pathways important for osteoclastogenesis in the CaPO_4_ and AngII mouse models, we are interested in examining the relative importance of TRAF2/TRAF6 signaling in human tissues. This may give us insight as to the relative contribution of the RANK/RANKL and CaPO_4_ + TNFα osteoclastogenesis pathways in human aneurysmal disease, and thus inform future therapeutic interventions. These experiments are forthcoming. Future studies will examine multiple potential therapeutic targets in the axis of OLC activation as it relates to aneurysm formation. These therapies will target OLC activation at the levels of stimulus, signaling, and effector function through the application of neutralizing antibodies, HIF-1α inhibitors, and protease inhibitors or bisphosphonates, respectively. It is apparent that many human diseases are multifactorial in their etiology or progression, therefore, we hypothesize that combination therapies, incorporating multiple targets in the OLC activation axis, will be most effective for the prevention or treatment of human AAAs.

## 4. Materials and Methods

### 4.1. Cell Culture and Treatments

The RAW 264.7 mouse macrophage cell line was purchased from the American Type Culture Collection (ATCC, Manassas, VA, USA) and maintained in Dulbecco’s modified Eagle’s medium (DMEM) (Mediatech, Manassas, VA, USA) containing 10% fetal bovine serum (FBS) (Mediatech), 100 IU/mL penicillin, and 100 μg/mL streptomycin (Mediatech). For macrophage stimulation, macrophages were maintained in modified Eagle’s medium-alpha (MEM-α) (Mediatech) supplemented with 10% charcoal-stripped FBS (Mediatech), penicillin, and streptomycin, and stimulated with or without 100 ng/mL TNFα (Peprotech, Rocky Hill, NJ, USA) plus 2% CaPO_4_ (*v*/*v*). The CaPO_4_ crystals were prepared by a 1:10 dilution of 1M CaCl_2_ (Sigma-Aldrich, St. Louis, MO, USA) to phosphate-buffered saline (PBS) (Sigma-Aldrich).

### 4.2. Human Tissue

Aneurysmal tissues were obtained from patients with a diagnosis of AAA, and an indication of maximum aortic diameter exceeding 5.5 cm, undergoing elective open repair surgery. Carotid plaque tissues were obtained from patients undergoing endarterectomy. All surgeries were performed at the University of Wisconsin Hospital. All normally discarded tissue obtained for this study was approved for use by the Minimal Risk-Health Sciences Institutional Review Board at the University of Wisconsin-Madison (submission ID 2018-0327, 25 April 2018). This tissue is not subject to IRB review because, in accordance with federal regulations, this project does not involve human subjects as defined under 45 CFR 46.102(e)(1).

### 4.3. Mouse Arterial Aneurysm Models and Treatments

Ten-week-old male C57BL/6 mice were obtained from the Jackson Laboratory (Bar Harbor, ME, USA). The procedures for creating our CaPO_4_-induced mouse model of arterial aneurysm were previously described [64]. Briefly, 0.5 M CaCl_2_-soaked gauze was applied perivascularly for 20 min on the carotid artery, as indicated. The gauze was replaced with another PBS-soaked gauze for 10 min, and the incised area was sutured. Untreated contralateral arteries served as controls. The mice were sacrificed seven days after surgery. Treated and control arteries were measured with an electronic digital caliper (VWR International, West Chester, PA, USA). Arteries collected for histological examinations were fixed by perfusion with 4% paraformaldehyde. To obtain live cells for flow cytometry, the arteries were not fixed, but perfused with DMEM prior to resection. AngII-induced aneurysms were produced in retired male breeder apoE^−/−^ mice (>6 months of age) obtained from the Jackson Laboratory, as previously described [22]. Mice received either Ang II (1000 ng/min/kg) (Sigma-Aldrich) or PBS via continuous infusion via micro-osmotic pump (Alzet model 1004, Durect Corporation, Cupertino, CA, USA) implanted in the back of the mouse. Mice were sacrificed 28 days post-implantation. Aortae collected for histological examinations were fixed by perfusion with 4% paraformaldehyde (Santa Cruz Biotechnology, Dallas, TX, USA). To obtain live cells for flow cytometry, the aortae were perfused with DMEM prior to resection. The maximum diameter of the abdominal aortae was measured ex vivo with digital calipers to evaluate aneurysm formation. All animal procedures were conducted in accordance with experimental protocols that were approved by the Institutional Animal Care and Use Committee at the University of Wisconsin-Madison (Protocol M005383, 8 April 2016).

### 4.4. Flow Cytometry

Mouse and human tissue samples were incubated at 37 °C with shaking for 1.5 h in digestion buffer containing 450 U/mL collagenase I (Sigma-Aldrich), 125 U/ml collagenase XI (Sigma-Aldrich), 60 U/mL hyaluronidase (Worthington Biochemical, Lakewood, NJ, USA), and 60 U/ml deoxyribonuclease I (Worthington), as previously described [65]. The digested tissue was then dissociated to obtain a single-cell suspension by passage through a 70 µm cell strainer (Falcon, Corning, NY, USA), and 2.5 × 10^5^ cells were stained for flow cytometric analysis as follows. First, the cells were blocked with 100 μg/mL of purified mouse IgG (Jackson Immuno-Research, West Grove, PA, USA) for 20 min at 4 °C. Then the cells were washed with wash buffer (PBS, 3% FBS) and stained for surface expression of CD11b and viability with anti-CD11b-BV711 antibody (563168, BD Biosciences, Franklin Lakes, NJ, USA) and Ghost 510 (Tonbo, San Diego, CA, USA), respectively, for 30 min at 4 °C. The cells were then washed and resuspended in fixation buffer (Invitrogen, San Diego, CA, USA), and allowed to fix for 15 min at room temperature. The cells were washed in permeabilization buffer (Invitrogen), and resuspended in permeabilization buffer containing the following primary antibodies: Anti-TRAP-Alexa Fluor 488 (ab216934, Abcam, Cambridge, UK), anti-cathepsin K-Alexa Fluor 647 (sc-48353, Santa Cruz Biotechnology), anti-MMP-9-PE (sc-13520, Santa Cruz Biotechnology), and anti-HIF-1α-Alexa 594 (sc-53546, Santa Cruz Biotechnology). The antibodies and cells were incubated for 30 min at room temperature. Following incubation, the cells were washed with permeabilization buffer and resuspended in wash buffer. The cells were analyzed on an Attune NxT Flow Cytometer (Invitrogen, Carlsbad, CA, USA). Fluorescence minus one controls were prepared and analyzed for each fluorochrome used.

### 4.5. Immunofluorescence Staining

Perfusion fixed arteries from mice treated with CaPO_4_ or AngII, and human carotid plaque and AAA tissue fixed overnight in 4% paraformaldehyde, were embedded with optimal cutting temperature compound, frozen, and sectioned at 8 μm. The sections were then rinsed in DI water and blocked with 5% donkey serum (Santa Cruz Biotechnology) for one hour. After blocking, the tissue sections were washed twice in 1X Tris-buffered saline-Tween 20 (TBST) (Affymetrix, Cleveland, OH, USA) and surface-stained with anti-CD11b primary antibody (ab133357, Abcam) overnight at 4 °C. After washing with TBST, the sections were incubated with donkey anti-rabbit Alexa Fluor 594 secondary antibody for one hour (Life Technologies, San Diego, CA, USA). After washing, the sections were permeabilized for 20 min in 10× TBST for intracellular staining. The sections were then washed and blocked with 1% bovine serum albumin (BSA) (Santa Cruz Biotechnology) for one hour. After washing, the sections were incubated overnight with anti-TRAP Alexa Fluor 488 antibody (ab216934, Abcam) at 4 °C. After overnight incubation, the sections were washed and counterstained with 4,6-diamidino-2-phenylindole dihydrochloride (DAPI) (Life Technologies) for 20 min. The sections were washed for a final time and mounted with Vectashield hardset antifade mounting medium (Vector Labs, Burlingame, CA, USA). The stained tissue sections were then examined with a Nikon E600 fluorescent microscope (Nikon, Tokyo, Japan).

### 4.6. Statistical Analysis

Data are reported as the means ± standard error of the means (SEM). The means of two groups were compared using the unpaired, two-tailed Student’s *t*-test. Statistical analysis was performed with the GraphPad Prism program, version 7.0 (GraphPad Software. Inc. San Diego, CA, USA). *p*-values < 0.05 were accepted as statistically significant.

## Figures and Tables

**Figure 1 ijms-20-04689-f001:**
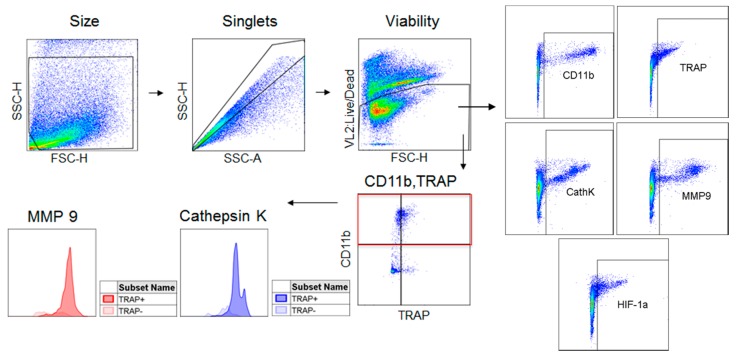
Flow cytometry gating strategy. Cells derived from culture or tissue samples were gated for size, singlets, and viability. Live cells were then analyzed for their expression of CD11b, TRAP, cathepsin K, MMP-9, and HIF-1α. The CD11b^+^ population was divided into TRAP^−^ and TRAP^+^ populations which were analyzed for their expression of cathepsin K and MMP-9.

**Figure 2 ijms-20-04689-f002:**
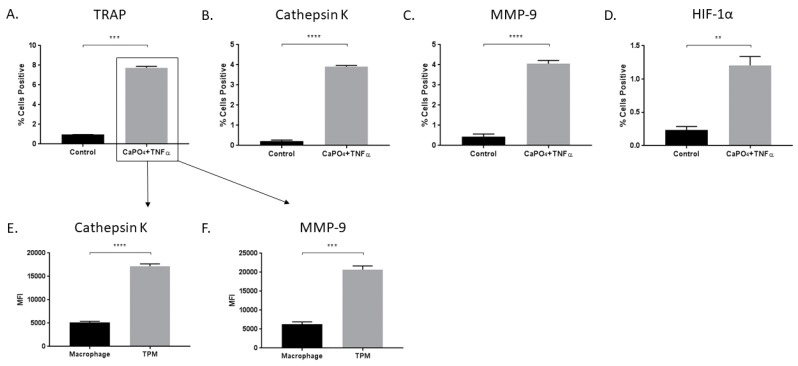
Treatment of macrophages with CaPO_4_ and TNFα results in osteoclastogenesis and increased protease production. RAW 264.7 macrophages were treated with CaPO_4_ and TNFα and analyzed for the percent of live cells expressing TRAP (**A**), cathepsin K (**B**), MMP-9 (**C**), and HIF-1α (**D**). (**E**,**F**) The expression (MFI) of cathepsin K and MMP-9, respectively, in TRAP^+^ cells (TPM) compared to TRAP^−^ cells (macrophage). The data represent three independent experiments. Data are expressed as mean ± SEM. ** *p* < 0.01, *** *p* < 0.001, **** *p* < 0.0001. MFI, median fluorescence intensity.

**Figure 3 ijms-20-04689-f003:**
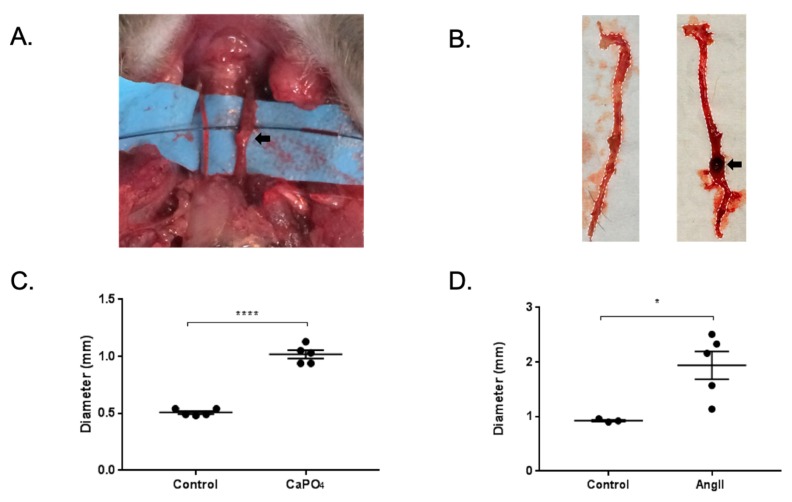
Mouse CaPO_4_ and AngII-induced aneurysms. (**A**,**B**) Representative images of mice treated via perivascular application of CaPO_4_ or subcutaneous administration of AngII, respectively. Black arrows indicate aneurysmal vessels. (**C**,**D**) CaPO_4_ and AngII treatment resulted in significant vessel dilation compared to control vessels (*n* = 5). In the CaPO_4_ model (**C**), the contralateral carotid arteries served as controls (*n* = 5), and PBS-only treated mice served as controls for the AngII model (*n* = 3) (**D**). These data are expressed as mean ± SEM. * *p* < 0.05, **** *p* < 0.0001.

**Figure 4 ijms-20-04689-f004:**
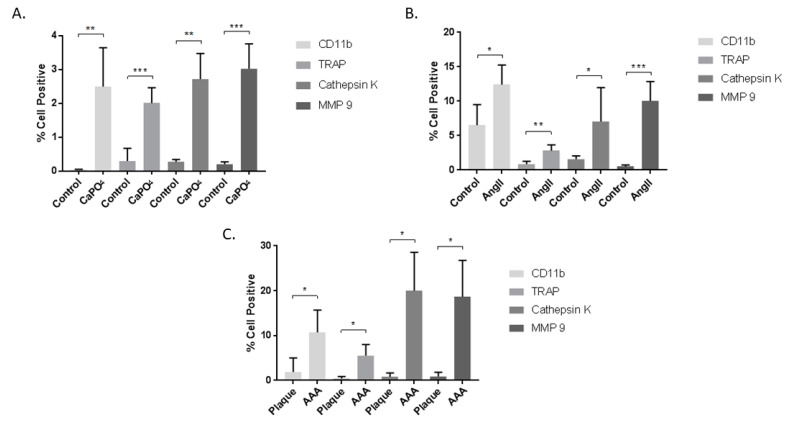
Cells from mouse CaPO_4_ (**A**), mouse AngII (**B**), and human (**C**) aneurysms exhibit increased expression of the myeloid marker, CD11b, and the osteoclastogenic markers TRAP, cathepsin K, and MMP-9. Tissues from CaPO_4_ mice (*n* = 5), AngII mice (*n* = 6) and PBS-only controls (*n* = 3), human AAA (*n* = 4), and human carotid plaques (*n* = 3) were enzymatically digested and processed to obtain single cell suspensions for flow cytometric analysis, as described in the Methods section. Data are expressed as mean ± SEM. * *p* < 0.05, ** *p* < 0.01, *** *p* < 0.001. AAA, abdominal aortic aneurysm.

**Figure 5 ijms-20-04689-f005:**
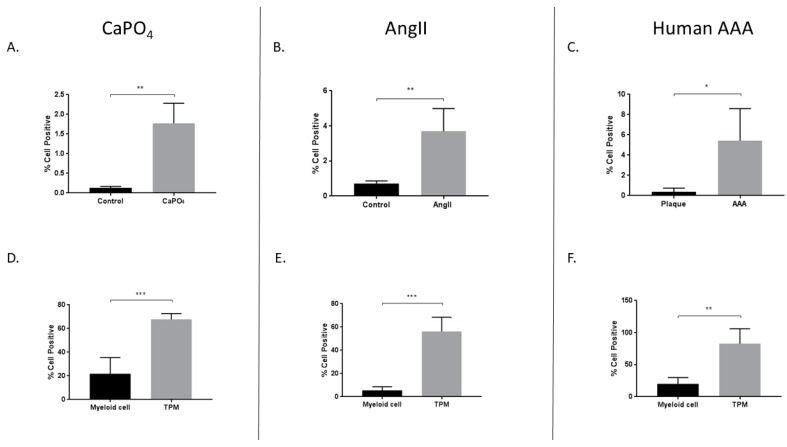
Increased expression of HIF-1α in mouse CaPO_4_ (**A**,**D**), mouse AngII (**B**,**E**), and human (**C**,**F**) aneurysms. Tissues from CaPO_4_ mice (*n* = 5), AngII mice (*n* = 5) and PBS-only controls (*n* = 3), human AAA (*n* = 4), and human carotid plaques (*n* = 3) were enzymatically digested and processed to obtain single cell suspensions for flow cytometric analysis, as described in the Methods section. (**A**–**C**) depict the percentage of live cells expressing HIF-1α in aneurysmal and control tissues. (**D**–**F**) compare the percentage of live cells expressing HIF-1α between myeloid cells (CD11b^+^, TRAP^−^) and TPMs (CD11b^+^, TRAP^+^) from aneurysmal tissues. Data are expressed as mean ± SEM. * *p* < 0.05, ** *p* < 0.01, *** *p* < 0.001. TPM, TRAP-positive macrophage. AAA, abdominal aortic aneurysm.

**Figure 6 ijms-20-04689-f006:**
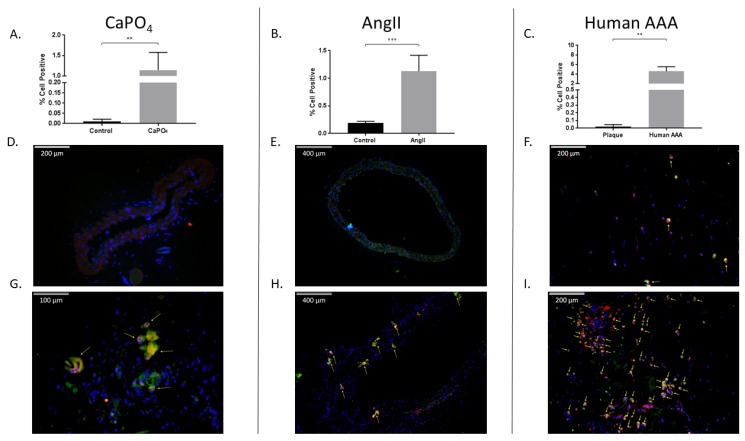
Mouse and human aneurysms contain more TPMs than non-aneurysmal control tissues. Mouse CaPO_4_ (**A**), mouse AngII (**B**), and human (**C**) aneurysms exhibit more TPMs (cells double-positive for expression of the myeloid marker, CD11b, and the osteoclast marker, TRAP) than respective controls. (**A**–**C**) show TPMs as a percentage of live cells in tissues from CaPO_4_ mice (*n* = 5) (**A**), AngII mice (*n* = 6) and PBS-only controls (*n* = 3) (**B**), human AAA (*n* = 4) and human carotid plaques (*n* = 3) (**C**). Tissues were enzymatically digested and processed to obtain single-cell suspensions for flow cytometric analysis, as described in the Methods section. (**D**–**I**) Immunofluorescence staining of frozen control and aneurysmal sections. Control sections (**D**–**F**), and aneurysmal sections (**G**–**I**) were stained for CD11b (red), TRAP (green), and DAPI (blue) as described in the Methods section. Yellow arrows indicate TPMs. 100× magnification (**E**,**H**), 200× magnification (**D**,**F**,**I**), 400× magnification (**G**). Data are expressed as mean ± SEM. ** *p* < 0.01, *** *p* < 0.001. TPM, TRAP-positive macrophage. AAA, abdominal aortic aneurysm.

**Figure 7 ijms-20-04689-f007:**
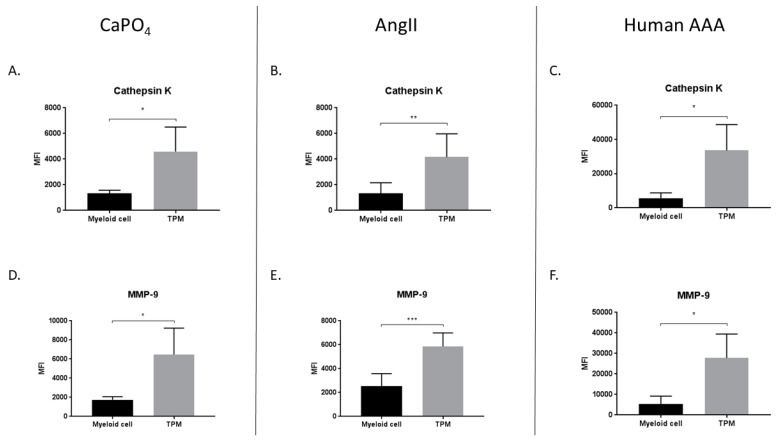
TRAP-positive macrophages produce more proteases than myeloid cells in mouse CaPO_4_ (**A**,**D**), mouse AngII (**B**,**E**), and human (**C**,**F**) aneurysms. Aneurysmal tissues from CaPO_4_-treated mice (*n* = 4), AngII-treated mice (*n* = 6), and human AAA (*n* = 4) were enzymatically digested and processed to obtain single-cell suspensions for flow cytometric analysis, as described in the Methods section. The expression of cathepsin K (**A**–**C**) and MMP-9 (**D**–**F**) is compared between myeloid cells (CD11b^+^, TRAP^−^) and TPMs (CD11b^+^, TRAP^+^). Data are expressed as mean ± SEM. * *p* < 0.05, ** *p* < 0.01, *** *p* < 0.001. TPM, TRAP-positive macrophage. AAA, abdominal aortic aneurysm.

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
