# Peer review of "Osteoclast-Like Cells in Aneurysmal Disease Exhibit an Enhanced Proteolytic Phenotype"

_ijms, 2019, doi:10.3390/ijms20194689_

Round 1
Reviewer 1 Report
Page 3, line 27: The diameters in the text (1.58 + 0.2773 mm vs 7.05 + 2.008 mm) are different from those seen in Figure 3D. These numbers are obviously not diameters but rather cathepsin K expression levels in aneurysmal tissues from the mouse AngII-induced model. Page 10, line 21: It is not only aortic but carotid tissue as well. A modification to the 4.3 paragraph should be made to describe the CaPO4 model, not the conventional CaCl2.Author Response
Response to Reviewer 1 comments
Point 1: Page 3, line 27: The diameters in the text (1.58 + 0.2773 mm vs 7.05 + 2.008 mm) are different from those seen in Figure 3D. These numbers are obviously not diameters but rather cathepsin K expression levels in aneurysmal tissues from the mouse AngII-induced model.
Response 1: Thank you for pointing out this error. The numbers have been corrected to reflect the diameters seen in Figure 3D. Please find the corrected numbers on Page 4, line 14.
Point 2: Page 10, line 21: It is not only aortic but carotid tissue as well.
Response 2: Thank you for pointing out this error. The word "aortic" has been removed to correct this error. Please find the corrected language on Page 10, line 31.
Point 3: A modification to the 4.3 paragraph should be made to describe the CaPO4 model, not the conventional CaCl2.
Response 3: Thank you for alerting us to this mistake. We have replaced "modified-CaCl2" with "CaPO4". Please find the corrected language on Page 10, line 39.
Reviewer 2 Report
Comments and Suggestions for Authors
The uptake of (18)F-fluorodeoxyglucose (FDG) in the aneurysm wall may be connected to the accumulation of macrophages and osteoclast-like cells responsible for the production and the activation of degrading enzymes [1]. Truijers et al. [2] reported that FDG uptake is located within the non-calcified aneurysm wall. This indirectly confirms that transfer of inflammatory cells and subsequently proteases such as matrix metalloproteinase-9 and cathepsin K occurs from the lumen through thin intraluminal thrombus to the underlying wall to a greater extent than from the vasa vasorum of the human AAA [3]. The mean annual growth rate was significantly lower in men with an AAA wall calcification above than below 50% [4]. AAA wall thickness varied regionally – from low at a rupture site to high at a calcified site [5]. In my opinion, the calcified plaques may be a significant barrier for the transport of inflammatory cells to the wall of the AAA. The development of heterogeneous multilayered intraluminal thrombi and calcified plaques likely has an effect on the thickness variability of the vascular wall. In this context, it has been suggested that the rupture site may be located within the non-calcified segment of the aneurysm wall.
Sakalihasan N, Hustinx R, Limet R. Contribution of PET scanning to the evaluation of abdominal aortic aneurysm. Semin Vasc Surg 2004;17:144-53. Truijers M, Kurvers HA, Bredie SJ, Oyen WJ, Blankensteijn JD. In vivo imaging of abdominal aortic aneurysms: increased FDG uptake suggests inflammation in the aneurysm wall. J Endovasc Ther 2008;15:462-7. Wiernicki I, Parafiniuk M, Kolasa-Wołosiuk A, Gutowska I, Kazimierczak A, Clark J, Baranowska-Bosiacka I, Szumilowicz P, Gutowski P. Relationship between aortic wall oxidative stress/proteolytic enzyme expression and intraluminal thrombus thickness indicates a novel pathomechanism in the progression of human abdominal aortic aneurysm. FASEB J. 2019;33:885-95. Lindholt JS. Aneurysmal wall calcification predicts natural history of small abdominal aortic aneurysms. Atherosclerosis 2008;197:673-8. Raghavan ML, Kratzberg J, Castro de Tolosa EM, Hanaoka MM, Walker P, da Silva ES. Regional distribution of wall thickness and failure properties of human abdominal aortic aneurysm. J Biomech 2006;39:3010-6.
Author Response
Response to Reviewer 2
Point 1: In my opinion, the calcified plaques may be a significant barrier for the transport of inflammatory cells to the wall of the AAA. The development of heterogeneous multilayered intraluminal thrombi and calcified plaques likely has an effect on the thickness variability of the vascular wall. In this context, it has been suggested that the rupture site may be located within the non-calcified segment of the aneurysm wall.
Response 1: Thank you for pointing out these additional studies that may emphasize the potential importance of osteoclast-like cells in the pathology of aneurysms. We have added this important information to the Introduction section on Page 1, lines 42 and 43, and Page 2, lines 21-23.
Reviewer 3 Report
This is a nice paper providing correlative data comparing aneurysmal tissue to non aneurysmal tissue. Unfortunately there is no pathway described. There is no proof that the findings are responsible for aneurysm development or growth. And there is no proof that the inhibition of the targets will avoid aneurysmal development or proof.
Author Response
Response to Reviewer 3
Point 1: There is no pathway described
Response 1: Thank you for pointing out the importance of describing the pathway involved in osteoclastogenesis in the context of aneurysm development. Our previous studies have examined osteoclastogenic signaling in the mouse AngII and CaPO4 models. Therefore, we have added discussion of these pathways to the Introduction section on Page 2, lines 11-15. We also added mention of the inhibition of osteoclastogenesis via HIF-1alpha inhibition in our previous work, which also demonstrates the potential importance of this signalling pathway in aneurysms. This addition can be found on Page 2, lines 40-42. Furthermore, we agree that a detailed examination of the pathways involved in stimulation of OLCs in vivo is vital, therefore we have included the absence of these experiments as a limitation of this study in the Discussion section, Page 10, lines 7-13.
Point 2: There is no proof that the findings are responsible for aneurysm development or growth
Response 2: Thank you for pointing out this important consideration. In the Introduction section we emphasized the importance of proteases, such as MMP-9, in aneurysm development. This addition can be found on Page 2, lines 21-25. We also outlined the important role of MMP-9 and cathepsin K in the first paragraph of the Discussion section, in which previous studies demonstrate that KO of either protease results in decreased aneurysm formation. Our results add to this discussion through elucidation of a specific cell population (i.e., OLCs) that are responsible for production of both MMP-9 and cathepsin K. Furthermore, we demonstrated that this unique cell population is present in both mouse and human aneurysmal tissues, thus pointing to OLCs as a potential theraputic target.
Point 3: There is no proof that the inhibition of the targets will avoid aneurysmal development
Response 3: Thank you for your critique indicating the importance of conducting experiments involving inhibition of OLCs. We added discussion of our previous OLC inhibition experiments in the mouse AngII and CaPO4 models to the Introduction section on Page 2, lines 25-30. We absolutely agree that these experiments are important, thus, we have included the absence of these experiments as a limitation of the paper in the Discussion section, Page 10 lines 6 and 7.